# Neural-Symbolic Architectural Axioms of Integration: A Manifesto

**Connor Pryor**                                           CONNOR.PRYOR@CAPITALONE.COM
**Lise Getoor**                                                          GETOOR@UCSC.EDU

**Editors:** Leilani H. Gilpin, Eleonora Giunchiglia, Pascal Hitzler, and Emile van Krieken

## Abstract

The integration of neural and symbolic methods has long been viewed as a promising path toward more general, interpretable, and robust artificial intelligence. The past two decades have seen a rapid proliferation of neural-symbolic (NeSy) systems, spanning a wide range of architectures, reasoning strategies, and application domains (Besold et al., 2022; d'Avila Garcez et al., 2019, 2002; Marra et al., 2024). However, this growth has outpaced theoretical clarity: many existing approaches conflate the roles of learning, inference, and representation, leading to a fragmented field lacking principled foundations. In this work, we address this gap by proposing a set of *architectural axioms of integration*—formal, implementation-agnostic principles that define how neural and symbolic components can be coherently combined. These axioms abstract away from system-specific details and instead characterize the structural interface between neural perception and symbolic reasoning. Rather than introducing a new method, this work offers a foundation to organize, compare, and reason about the rapidly expanding space of NeSy approaches.

## 1. Introduction

The promise of integrating neural and symbolic methods to leverage their complementary strengths has long been a motivating vision in artificial intelligence. Neural-symbolic (NeSy) AI has emerged from this aspiration, growing steadily over the past two decades with regular workshops since 2005 (NeSy2005) and culminating in its first dedicated conference in 2024 (NeSy2024). At its core, NeSy research seeks to develop models and algorithms that systematically combine the statistical generalization capabilities of neural networks with the compositional, interpretable, and often discrete structure of symbolic formalisms (Ahmed et al., 2022a; Badreddine et al., 2022; Cohen et al., 2020; d'Avila Garcez et al., 2019, 2002, 2009; Manhaeve et al., 2021; Pryor et al., 2023; Xu et al., 2018; Yang et al., 2020)

The intellectual roots of NeSy AI lie in the long-standing divide between symbolic and connectionist approaches, which became particularly pronounced during the rise of neural networks in the 1980s and 1990s. Patrick Winston (1991) captured this dichotomy and the potential for synthesis in his reflection:

> "Today, some researchers who seek a simple, compact explanation hope that systems modeled on neural nets or some other connectionist idea will quickly overtake more traditional systems based on symbol manipulation. Others believe that symbol manipulation, with a history that goes back millennia, remains the only viable approach. ... Instead, ... AI must use many approaches. AI is not like circuit theory and electromagnetism. There is nothing wonderfully unifying like Kirchhoff's laws are to circuit theory or Maxwell's equations are to electromagnetism. Instead of looking for a 'right way,' the time has come to build systems out of diverse components, some connectionist and some symbolic, each with its own diverse justification." (Minsky, 1991, page 35)

Despite this early recognition, symbolic approaches were largely eclipsed during the deep learning revolution and only in the past decade has neural-symbolic integration regained significant attention. This resurgence has led to a proliferation of methods, architectures, and loss functions (Agrawal et al., 2019; Ahmed et al., 2023a,b, 2022b,a; Amos and Kolter, 2017; Badreddine et al., 2023, 2022; Cingillioglu and Russo, 2019; Cohen et al., 2020; Cornelio et al., 2023; Cunnington et al., 2024; Dasaratha et al., 2023; Derkinderen et al., 2024; Diligenti et al., 2017a,b; Dong et al., 2019; Giunchiglia et al., 2022; Hu et al., 2016; Jung et al., 2024; Maene and Raedt, 2024; Manhaeve et al., 2021; Marra, 2022; Marra and Kuželka, 2021; Marra et al., 2019, 2020; Martires et al., 2024; Misino et al., 2022; Pan et al., 2023; Pryor et al., 2023; Rocktäschel and Riedel, 2017; Scarselli et al., 2009; Serafini and d'Avila Garcez, 2016; Shindo et al., 2023; Sikka et al., 2020; Sourek et al., 2018; Stoian et al., 2023; Towell and Shavlik, 1994; Tran and d'Avila Garcez, 2018; van Krieken et al., 2023; Wang et al., 2019; Winters et al., 2022; Xu et al., 2018; Yang et al., 2017, 2020).

However, the field's rapid expansion has outpaced its theoretical consolidation. Much of NeSy research from the mid-2010s onward has followed a systems-first approach: extending or hybridizing existing platforms (e.g., ProbLog, Answer Set Programs, PSL, MLNs, circuits, etc.) to support neural components. This has led to a conceptual ambiguity about what constitutes a neural-symbolic system and even how each system differs from one another. For some, symbolic knowledge must be represented using formalisms with well-defined syntax and semantics (e.g., logic or probabilistic logic); for others, symbolic structure may be interpreted more broadly to include natural language, graph structures, or program traces. As a result, NeSy remains an early-stage, fragmented field with no universally accepted theoretical grounding.

To address this ambiguity, the community has proposed a range of taxonomies that classify NeSy systems along multiple dimensions, including representation of symbolic knowledge (Besold et al., 2022; van Krieken, 2024; Marra et al., 2024; van Krieken et al., 2022), the interaction between neural and symbolic components (Bader and Hitzler, 2005; Besold et al., 2022; d'Avila Garcez et al., 2019, 2009; Dash et al., 2022; Dickens, 2024; Dickens et al., 2024b; Marra et al., 2024; van Bekkum et al., 2021), learning and reasoning (d'Avila Garcez et al., 2019; Dickens, 2024; Dickens et al., 2024a; Marra et al., 2024; van Bekkum

et al., 2021), application domains (Badreddine et al., 2022; Bortolotti et al., 2025; Dickens, 2024; Manhaeve et al., 2021; Marra et al., 2024; Yu et al., 2023), common shortcuts (Marconato et al., 2023a, 2024, 2023b), and system languages (van Krieken et al., 2024). They also bridge connections to related fields such as graph neural networks (Lamb et al., 2020), statistical relational learning (Marra et al., 2024), and knowledge graphs (Zhang et al., 2021). While these taxonomies provide valuable structure and insight, they address only the symptoms of a larger underlying challenge: the need for principled foundations for neural-symbolic AI. Given the decades of separate neural and symbolic research, fundamentally, NeSy should begin with how to integrate these two sets of theories, i.e., it needs axioms of neural-symbolic integration. Such foundations are essential for unifying the field, fostering communication within and beyond NeSy, and preventing the redundant reinvention of ideas.

This work is not an empirical comparison, nor does it introduce a new neural-symbolic method or learning procedure. Instead, it addresses a foundational gap in the field by proposing a set of *architectural axioms of integration*. The goal is to abstract away from specific implementations and task-specific designs, and to focus on the underlying structural principles that govern integration itself. The proposed axioms are intended to serve as a theoretical starting point, i.e., a foundation upon which current and future neural-symbolic architectures, theoretical developments, and practical systems can be systematically understood, compared, and constructed.

## 2. Related Work

The question of how information flows, such as knowledge, variables, gradients, values, or probabilities, between neural and symbolic components lies at the heart of almost every neural-symbolic system (Agrawal et al., 2019; Ahmed et al., 2023a,b, 2022b,a; Amos and Kolter, 2017; Badreddine et al., 2023, 2022; Cingillioglu and Russo, 2019; Cohen et al., 2020; Cornelio et al., 2023; Cunnington et al., 2024; Dasaratha et al., 2023; Derkinderen et al., 2024; Diligenti et al., 2017a,b; Dong et al., 2019; Giunchiglia et al., 2022; Hu et al., 2016; Jung et al., 2024; Maene and Raedt, 2024; Manhaeve et al., 2021; Marra, 2022; Marra and Kuželka, 2021; Marra et al., 2019, 2020; Martires et al., 2024; Misino et al., 2022; Pan et al., 2023; Pryor et al., 2023; Rocktäschel and Riedel, 2017; Scarselli et al., 2009; Serafini and d'Avila Garcez, 2016; Shindo et al., 2023; Sikka et al., 2020; Sourek et al., 2018; Stoian et al., 2023; Towell and Shavlik, 1994; Tran and d'Avila Garcez, 2018; van Krieken et al., 2023; Wang et al., 2019; Winters et al., 2022; Xu et al., 2018; Yang et al., 2017, 2020). Although each of these works implicitly or explicitly defines a specific flow of information between neural and symbolic modules, comparatively little effort has been devoted to synthesizing these designs into a coherent, unifying theory of integration.

That said, several attempts have been made to categorize or formalize neural-symbolic systems through taxonomies, design patterns, or architectural diagrams. These efforts often use visual metaphors such as boxes and arrows to illustrate the interaction between neural

and symbolic components (Yu et al., 2023; van Bekkum et al., 2021), or tabulate systems based on representational or functional criteria (d'Avila Garcez et al., 2019; Marra et al., 2024). Such work provides valuable insight into the diversity of existing approaches, especially with respect to knowledge representations, inference types, and application domains. However, these approaches often conflate integration with inference or learning, focus narrowly on specific knowledge representations (usually tied to traditional symbolic systems), or remain overly abstract in their discussion of integration, making them less actionable. As a result, they often serve as patterns or design taxonomies rather than foundational principles of integration. In contrast, this work takes an orthogonal approach. We aim to go one level deeper by identifying core principles that describe how neural and symbolic components may coherently exchange and transform information. These axioms abstract away from any particular symbolic formalism or neural architecture, instead capturing the essential connections between the two theories.

## 3. Architectural Axioms of Integration

To systematically characterize the design space of neural-symbolic systems, we propose a set of **architectural axioms of integration**, principled abstractions that define how neural and symbolic components are structurally combined. These axioms are not tied to any specific learning or inference paradigm; rather, they describe the possible configurations and information pathways that enable integration, independent of how systems are trained, queried, or optimized. We organize the axioms into two broad categories: **catalyst** (Section 3.1) and **interface** (Section 3.2).

- **Catalyst axioms** (Section 3.1) characterize transformations that occur across representational boundaries. In these cases, one component, neural or symbolic, serves as a *constructive or deconstructive mechanism* for the other. For example, symbolic knowledge may be used to construct the architecture of a neural network, or conversely, trained neural models may be deconstructed to extract symbolic structures. Catalyst-based integration captures these asymmetrical transformations, where one modality reshapes or defines the other.

- **Interface axioms** (Section 3.2) describe mechanisms by which neural and symbolic components *interact within a unified system*, enabling the exchange of information such as variables, parameters, gradients, scores, rewards, etc. Unlike catalyst-based integration, interface-based does not imply transformation from one representation to another, but instead emphasizes *cooperation* across modalities. These interfaces govern how the two components communicate during execution, whether that be through differentiable pathways, sampling procedures, or latent representations.

Together, these two categories span the architectural core of neural-symbolic design. The following subsections articulate each axiom type in detail, together with an illustrative

NeSy system example (for an example of an interface-based NeSy task, see Appendix B). But first, let us introduce the notation that will be used throughout the rest of the paper.

**Notation.** Let $\mathbf{X}_{nn} = \{X_1^{nn}, \ldots, X_n^{nn}\}$ denote *observed random variables* processed by the neural architecture, with realizations $\mathbf{x}_{nn}$ representing known values (e.g., $X_i^{nn}$ might represent an image or sensor input). Let $\mathbf{X}_{sy} = \{X_1^{sy}, \ldots, X_p^{sy}\}$ denote observed symbolic variables (e.g., logical facts or structured inputs), with realizations $\mathbf{x}_{sy}$. We denote $\mathbf{Y} = \{Y_1, \ldots, Y_m\}$ as *target random variables* to be predicted, and $\mathbf{Z} = \{Z_1, \ldots, Z_k\}$ as *latent random variables* representing unobserved structure, with realizations $\mathbf{y}$ and $\mathbf{z}$, respectively. We denote *symbolic architectures* by $\phi_{sy}$, parameterized by weights $\mathbf{w}_{sy}$, and *neural architectures* by $g_{nn}$, parameterized by weights $\mathbf{w}_{nn}$.[1]

## 3.1. Catalyst Axioms

Catalyst-based integration refers to architectures in which one modality is used to construct or extract structure from the other. Unlike interface-based integration, which captures the exchange of information during model execution, catalyst-based methods define a directional architectural transformation: one component serves as a structural catalyst for the other. These transformations are structural rather than operational, i.e., they reshape the space of variables, functions, or model architecture rather than simply passing values or gradients.

We identify three primary forms of catalyst-based integration. In *direct neural construction* (Section 3.1.1), symbolic knowledge is directly translated as the architecture of a neural network. In *programmatic neural construction* (Section 3.1.2), symbolic structures are used to algorithmically generate neural models through code or templates. In *symbolic extraction* (Section 3.1.3), symbolic representations are derived from neural networks, enabling interpretation, rule induction, or verification.

### 3.1.1. Direct Neural Construction

*Direct neural construction* refers to architectures in which symbolic structure is directly mapped onto the components of a neural model, defining the topology or connectivity of the network. In this setting, the symbolic knowledge encoded in a model $\phi_{sy}$ serves as an architectural blueprint for the neural network $g_{nn}$, such that the symbolic relations and variables in $\phi_{sy}$ are deterministically instantiated as neurons, edges, or modules within $g_{nn}$. The symbolic component does not interact with the neural model during execution; instead, it provides a static, one-time specification of the neural architecture. More formally, a direct construction procedure defines a neural model $g_{nn}(\mathbf{x}_{nn}, \mathbf{y}; \mathbf{w}_{nn})$ whose topology is derived from the symbolic structure of $\phi_{sy}$. In such constructions, neurons may correspond

---

1. While many symbolic systems incorporate weights (e.g., probabilistic logic, soft logic), this is not universally the case. In purely logical or rule-based systems, such as propositional or first-order logic, $\phi_{sy}$ may have no associated weights, i.e., $\mathbf{w}_{sy} = \emptyset$.

to symbolic random variables (e.g., from $\mathbf{X}_{sy}, \mathbf{Y}, \mathbf{Z}$), and connections encode logical or relational dependencies specified by $\phi_{sy}$. While the neural parameters $\mathbf{w}_{nn}$ remain learnable, the architecture of $g_{nn}$ is fixed at design time by the symbolic model.

**Example 1 (Knowledge-Based Artificial Neural Networks)** *KBANN (Towell and Shavlik, 1994) exemplifies direct neural construction by translating symbolic knowledge bases into neural network architectures. In KBANN, a propositional rule base defined over symbolic variables is compiled into a layered feedforward neural network. The transformation relies on the following symbolic-to-neural correspondences:*

| *Knowledge Base Element* | *Neural Network Component* |
|---|---|
| *Final conclusions (i.e., $\mathbf{y}$)* | *Output units (i.e., $\mathbf{y}$)* |
| *Supporting facts (i.e., $\mathbf{x}_{sy}$)* | *Input units (i.e., $\mathbf{x}_{nn}$)* |
| *Intermediate conclusions (i.e., $\mathbf{z}$)* | *Hidden units (i.e., $\mathbf{z}$)* |
| *Dependencies (e.g., $A \wedge B \Rightarrow C$)* | *Weighted connections (i.e., $\mathbf{w}_{nn}$)* |

*Each logical rule (e.g., $A \wedge B \Rightarrow C$) is translated into a hidden unit with incoming connections from neurons corresponding to $A$ and $B$, and an outgoing connection to $C$.*

Direct neural construction introduces symbolic inductive bias directly into the structure of the neural network. This promotes modularity and interpretability while constraining the hypothesis space to reflect domain knowledge. However, the approach is limited by the expressiveness and completeness of the symbolic model: if $\phi_{sy}$ is sparse, inconsistent, or ambiguous, the resulting neural network may lack the flexibility required for generalization.

### 3.1.2. PROGRAMMATIC NEURAL CONSTRUCTION

*Programmatic neural construction* refers to architectures in which symbolic structure is used to programmatically instantiate a neural model. In contrast to direct neural construction, which enforces a static mapping between symbolic variables and neural components, programmatic construction interprets the symbolic model $\phi_{sy}$ as a generative specification for assembling a neural architecture $g_{nn}$. This process encodes symbolic *inductive bias* (i.e., assumptions about structure, modularity, or invariances) directly into the network's topology or parameterization. More formally, a compilation process interprets $\phi_{sy}$ to generate a neural model $g_{nn}(\mathbf{x}_{nn}, \mathbf{y}; \mathbf{w}_{nn})$, where symbolic constraints determine how components are selected, arranged, or connected. The resulting architecture reflects the structure of $\phi_{sy}$, even though no symbolic inference is performed at runtime.

**Example 2 (CNNs as Programmatic Inductive Bias)** *Convolutional neural networks (CNNs) can be interpreted as a form of programmatic neural construction. Though commonly considered purely neural models, CNNs encode an architectural prior over the input*

*domain: they assume spatial locality and translation invariance. These assumptions are enforced programmatically by restricting connections to spatially adjacent inputs and applying shared filters across locations.*

While this interpretation casts programmatic neural construction as a general mechanism for imposing inductive bias, it also raises an important caveat. Given this generously broad framing, many neural architectures, such as convolutional, recurrent, and transformer models, could be viewed as programmatic neural constructions. In practice, NeSy systems that fall under programmatic construction typically exhibit a more explicit and semantically grounded role for symbolic structure. For example, in structured classification tasks such as MNIST-Addition, symbolic programs are used to assemble neural components by instantiating and composing digit classifiers in a way that reflects the semantic structure of the problem—e.g., aligning network modules with digit positions in an arithmetic expression.

### 3.1.3. Symbolic Extraction

*Symbolic extraction* refers to architectures in which symbolic representations are derived from trained neural models. In contrast to construction-based approaches that embed symbolic structure into neural architectures, extraction-based catalysts operate in the reverse direction: a neural model $g_{nn}(\mathbf{x}_{nn}, \mathbf{y}; \mathbf{w}_{nn})$ is treated as a trained, potentially opaque function, and symbolic knowledge is induced by analyzing its learned behavior or internal representations. The resulting symbolic output may take the form of logical rules, decision trees, algebraic formulas, or other interpretable structures. More formally, symbolic extraction aims to derive a symbolic model $\phi_{sy}$ that approximates or explains the behavior of $g_{nn}$. The symbolic representation may capture global decision boundaries or be restricted to local input regions of interest. In either case, $\phi_{sy}$ is constructed post hoc and is not used during training, but instead serves to interpret or verify the neural model's learned behavior.

**Example 3 (Distilling Neural Networks into Soft Decision Trees)** *Frosst and Hinton (2017) propose a method for distilling a deep neural network into a soft decision tree. The soft decision tree is trained to mimic the output distribution of a teacher network $g_{nn}$ by minimizing the KL divergence between their predictive distributions over $\mathbf{y}$. Internal nodes in the tree compute soft decisions using logistic functions, and the symbolic structure emerges as a differentiable tree where each path encodes a symbolic decision rule. The resulting symbolic model $\phi_{sy}$ approximates $g_{nn}$ while remaining human-interpretable and analytically tractable.*

### 3.2. Interface Axioms

Interface-based integration refers to architectures in which neural and symbolic components interact during execution, exchanging information such as variable values, gradients, pa-

rameters, or structured representations. Unlike catalyst-based methods, where one modality constructs or distills the other, interface-based methods emphasize *ongoing interaction* between components, often during both inference and learning. These systems maintain a persistent coupling between the neural network $g_{nn}$ and symbolic module $\phi_{sy}$.

We identify three primary forms of interface-based integration. In a *variable interface* (Section 3.2.2), neural and symbolic components exchange values and gradients/rewards over shared random variables. In a *parameter interface* (Section 3.2.3), neural and symbolic components exchange values and gradients/rewards over shared parameters. In a *structure interface* (Section 3.2.4), the symbolic structure itself is dynamically defined by the neural network.

### 3.2.1. GRADIENT-BASED VS. SAMPLING-BASED INTERFACES

A foundational design choice in interface-based integration is whether the symbolic component $\phi_{sy}$ is differentiable with respect to the neural component $g_{nn}$. In a *gradient-based interface*, $\phi_{sy}$ is either inherently differentiable or relaxed into a continuous approximation that permits gradients to flow from symbolic losses into the neural parameters $\mathbf{w}_{nn}$. This allows for end-to-end training using standard backpropagation and is commonly employed in systems using fuzzy logic (Badreddine et al., 2022), soft constraints (Pryor et al., 2023), or probabilistic relaxations (Manhaeve et al., 2021; Xu et al., 2018). In contrast, a *sampling-based interface* is required when $\phi_{sy}$ is non-differentiable, such as in discrete or combinatorial symbolic models. In these cases, optimization proceeds through sampling-based methods such as reinforcement learning, rejection sampling, or black-box likelihood estimation, where neural updates are guided indirectly via rewards or sampled outputs.

### 3.2.2. VARIABLE INTERFACE

A *variable interface* refers to architectures in which the neural network $g_{nn}$ and symbolic model $\phi_{sy}$ are coupled through a shared set of random variables, enabling joint reasoning over latent, observed, or target quantities. In this formulation, the output of the neural model defines or conditions variables that are used as inputs to the symbolic model, or conversely, the output or latent variables of the symbolic model influence the input or latent states of the neural component. More formally, a variable interface connects $g_{nn}$ and $\phi$ through a shared subset of random variables $\mathbf{x}_{nn}$, $\mathbf{x}_{sy}$, $\mathbf{y}$, and $\mathbf{z}$. For example, the neural model may produce predictions over a subset of variables $\mathbf{v} \subseteq \{\mathbf{x}_{sy}, \mathbf{z}\}$, which are then used as inputs to $\phi_{sy}$. Alternatively, $\phi_{sy}$ may define constraints or latent dependencies over variables $\mathbf{v}' \subseteq \{\mathbf{x}_{nn}, \mathbf{z}\}$ that are in turn passed to or conditioned upon by $g_{nn}$.

**Example 4 (Neural-Symbolic Fuzzy Logic)** *A notable implementation of the variable interface arises in neural-symbolic fuzzy logic systems (Badreddine et al., 2022; Pryor et al., 2023), where neural models define truth values over a set of variables, and these values are interpreted within a differentiable fuzzy logic framework. Fuzzy logic generalizes classical*

*logic by allowing truth values to range continuously in $[0, 1]$, enabling reasoning over partial truths. Logical operators are replaced with differentiable relaxations—for example, conjunction $\wedge$ may be interpreted as multiplication, and negation $\neg A$ as $1 - A$. A fuzzy rule such as $A \wedge \neg B$ is thus approximated by the function $A \cdot (1 - B)$, where $A$ and $B$ are real-valued outputs from $g_{nn}$. Further details can be found in Appendix C.1.*

### 3.2.3. Parameter Interface

A *parameter interface* refers to architectures in which the neural network $g_{nn}$ and symbolic model $\phi_{sy}$ interact through the exchange of parameters that govern symbolic behavior. In this formulation, the neural model provides values that are interpreted as symbolic parameters, such as weights on facts, rules, or potentials, used by $\phi_{sy}$ to define distributions, loss functions, or structural constraints. Conversely, the symbolic model may also define parameters that condition neural inference, initialization, or representations. In both cases, the interface enables one component to modulate the functional behavior of the other via parameterization. More formally, a parameter interface arises when either (i) $\mathbf{w}_{sy}$ is defined as a function of the neural output, i.e., $\mathbf{w}_{sy} = g_{nn}(\mathbf{x}, \mathbf{y}; \mathbf{w}_{nn})$, or (ii) the symbolic component generates parameter constraints or priors over $\mathbf{w}_{nn}$.

**Example 5 (Neural-Symbolic Probabilistic Logic)** *A representative implementation of the parameter interface arises in neural-symbolic probabilistic logic systems (Manhaeve et al., 2021; Xu et al., 2018; Yang et al., 2020), where a neural network defines the probabilities over facts or rules in a probabilistic logic program. Probabilistic logic extends classical logic by associating probabilities with logical atoms or implications, thereby enabling reasoning under uncertainty. For example, a probabilistic fact such as "Alice smokes" with probability $0.7$ is represented as $0.7 :: Smokes("Alice")$. In the parameter interface setting, the probability $0.7$ is not manually specified but is instead predicted by a neural network conditioned on observed input $\mathbf{x}_{nn}$. Further details can be found in Appendix C.2.*

### 3.2.4. Structure Interface

A *structure interface* refers to architectures in which the neural model $g_{nn}$ is responsible for generating, selecting, or modifying the structure of the symbolic model $\phi_{sy}$. Unlike variable and parameter interfaces, where the symbolic component is fixed, structure interfaces allow the symbolic model itself to vary as a function of the neural output. The symbolic model is instantiated dynamically, often on a per-instance basis, such that each input may yield a different $\phi_{sy}$. More formally, the symbolic model $\phi_{sy}$ is generated directly by the neural model as a structured output: $\phi_{sy} \leftarrow g_{nn}(\mathbf{x}_{nn}, \mathbf{y}; \mathbf{w}_{nn})$. Since symbolic inference is generally non-differentiable, supervision is typically provided via reward signals or other forms of feedback rather than gradient-based updates.

**Example 6 (Structure Interface in LOGIC-LM)** *LOGIC-LM (Pan et al., 2023) exemplifies a structure interface by integrating a large language model (LLM) with a symbolic*

*solver to perform faithful logical reasoning. In this system, the LLM $g_{nn}(\mathbf{x}_{nn}, \mathbf{y}; \mathbf{w}_{nn})$ receives a natural language prompt $\mathbf{x}_{nn}$ and generates a symbolic formalization of the problem—typically in first-order logic—represented as a symbolic program $\phi_{sy}$. The resulting $\phi_{sy}$ is then executed independently by a deterministic symbolic solver to infer conclusions or verify logical consistency.*

## 4. Conclusion and Limitations

As the world shifts from merely exploring the potential of artificial intelligence to increasingly relying on its real-world applications, the urgency for responsible deployment and careful interpretation of model outputs has become profoundly evident. Neural-symbolic AI stands as a field uniquely positioned to address some of the most critical challenges in this transition, including the provision of interpretability, the enforcement of structured constraints, and the assurance of consistent and reliable predictions. However, despite its significant promise, NeSy AI remains a relatively nascent and fragmented field, often characterized by ad-hoc implementations lacking theoretical cohesion and standardization. As the importance of neural-symbolic methods grows, so too does the need for a principled and unified foundation to guide their development.

This work contributes toward establishing such a foundation by introducing a set of axioms for neural-symbolic integration. Rather than focusing on specific methods, learning procedures, or application domains, the axioms adopt an architectural perspective, identifying the fundamental design choices that govern how neural and symbolic components interact. This work categorizes integration into two primary modes: *catalyst-based* architectures, where one component influences the construction of the other, and *interface-based* architectures, where neural and symbolic modules are coupled through an explicit interface. By abstracting over particular models and domains, the proposed axioms offer a unifying lens through which existing systems can be understood, compared, and extended.

While the proposed axioms offer a principled starting point for neural-symbolic research, several limitations warrant acknowledgment. First, these axioms represent one possible abstraction among many; alternative perspectives, such as system-level taxonomies, may provide complementary or more expressive viewpoints. Second, while the framework captures a wide range of existing methods, it is not exhaustive. Some NeSy approaches may fall outside these current categories. For example, whether prompting LLMs constitutes a form of neural-symbolic integration and, if so, what axiom this should be classified as.

Looking ahead, we argue that one of the most pressing direction for future work is to build consensus around a shared set of foundational principles for neural-symbolic AI. Whether or not the specific formulation proposed here is adopted, a unifying theoretical foundation is necessary to reduce the overhead of describing new systems, facilitate collaboration across subfields, and foster clearer communication for new researchers entering the area. Without such a foundation, the field risks continued fragmentation and redundancy.

## 5. Acknowledgments

This work was supported in part by the National Science Foundation under Grant No. CCF-2023495. The first author is affiliated with the University of California, Santa Cruz, and Capital One. We are grateful to the many researchers who have contributed valuable discussions and insights over the years, including Charles Dickens, Eleonora Giunchiglia, Emile van Krieken, Jaron Maene, Lennert de Smet, Eriq Augustine, and many others whose input has helped shape this work.

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

## Appendix A. Introduction

This appendix provides additional technical and illustrative material that complements the main text. It includes a canonical NeSy example, as well as extended illustrations of a few of the architectural axioms defined in Section 3. The appendix is organized as follows:

- **Task Example** (Appendix B): A detailed walkthrough of a representative neural-symbolic task, illustrating the structure and dynamics of interface-based integration in NeSy systems.

- **Extended Axiom Examples** (Appendix C): Extended details on a few of the architectural axioms introduced in the main text.

## Appendix B. NeSy Task Example

This section presents a concrete neural-symbolic task designed to illustrate interface-based integration (Section 3.2), which has emerged as the central focus of neural-symbolic research in the past half-decade. No example has more clearly defined this direction than the MNIST-Addition task (Manhaeve et al., 2021), which exemplifies an interface-based architecture: the output of a neural network is passed as input to a symbolic model.

**Example 7 (MNIST-Addition)** *MNIST-Addition Manhaeve et al. (2021) is a canonical neural-symbolic problem involving the prediction of the sum of two digits, each represented as an MNIST image. The neural component $g_{nn}(\mathbf{x}_{nn}, \mathbf{y}, \mathbf{z}; \mathbf{w}_{nn})$, parameterized by weights $\mathbf{w}_{nn}$, serves as a digit classifier. It maps an input image $\mathbf{x}_{nn} \in \mathbb{R}^{28 \times 28}$ to a distribution over digit labels $\mathbf{y} \in \{0, \ldots, 9\}$. The symbolic component $\phi_{sy}$ encodes an arithmetic constraint representing addition over predicted digits.*

**Note B.1** *The precise formulation of a NeSy system depends on how neural predictions are integrated into symbolic inference, how the symbolic model $\phi_{sy}$ is defined, and the nature of the reasoning process.*

Suppose the task is to compute the sum $\mathbf{3} + \mathbf{5} = 8$. The two input images are passed through the neural model to produce predictive distributions:

$$g_{nn}(\mathbf{3}) = \{P(0)_1 = 0.0, P(1)_1 = 0.0, P(2)_1 = 0.0, P(3)_1 = 0.9, P(4)_1 = 0.0,$$
$$P(5)_1 = 0.0, P(6)_1 = 0.0, P(7)_1 = 0.0, P(8)_1 = 0.1, P(9)_1 = 0.0\},$$
$$g_{nn}(\mathbf{5}) = \{P(0)_2 = 0.0, P(1)_2 = 0.0, P(2)_2 = 0.0, P(3)_2 = 0.0, P(4)_2 = 0.0,$$
$$P(5)_2 = 0.8, P(6)_2 = 0.0, P(7)_2 = 0.0, P(8)_2 = 0.0, P(9)_2 = 0.2\}.$$

Assuming symbolic reasoning is performed over the most probable predictions (i.e., `argmax`), the symbolic component enforces the constraint:

$$\phi_{sy}\left(\arg\max g_{nn}(\mathbf{3}), \arg\max g_{nn}(\mathbf{5})\right) = 3 + 5 = 8.$$

**Note B.2** *Learning in this system may depend on whether ground-truth digit labels or the final sum is supervised, whether $\phi_{sy}$ is differentiable, and how symbolic reasoning is coupled to the neural model. When $\phi_{sy}$ is part of a differentiable system, gradients may flow from the output sum back into $g_{nn}$, enabling end-to-end training.*

## Appendix C. Extended Axiom Examples

This section provides a more detailed elaboration of select architectural axioms, focusing in particular on the *variable interface* (Section C.1) and *parameter interface* (Section C.2). These two forms of integration are currently among the most widely explored in the neural-symbolic community, due to their compatibility with differentiable learning frameworks and their applicability across a wide range of structured reasoning tasks. The extended examples presented here are intended to clarify the practical implementation of these interfaces, highlight their architectural significance, and illustrate how symbolic reasoning can be composed with neural perception in a unified system.

### C.1. Neural-Symbolic Fuzzy Logic

A notable example of the *variable interface-based* (Section 3.2.2) approach is fuzzy neural-symbolic systems, where neural models define observed variables that are embedded in a differentiable fuzzy logic framework (Badreddine et al., 2022; Pryor et al., 2023). Fuzzy logic generalizes classical logic by allowing truth values to range continuously over $[0, 1]$, enabling reasoning over partial truths. For example, a rule like $A \wedge \neg B$ can be approximated using differentiable operators as $A \cdot (1 - B)$.

**Note C.1** *NeSy fuzzy logic systems can be instantiated using various continuous-valued t-norms. Table 1 shows three commonly used in the NeSy literature: Gödel, Product, and*

| Logic | $\neg x$ | $x \wedge y$ | $x \vee y$ |
|---|---|---|---|
| Gödel | $1 - x$ | $\min(x, y)$ | $\max(x, y)$ |
| Product | $1 - x$ | $x \cdot y$ | $x + y - x \cdot y$ |
| Łukasiewicz | $1 - x$ | $\max(0, x + y - 1)$ | $\min(1, x + y)$ |

Table 1: Common fuzzy logic operations for negation, conjunction, and disjunction.

*Łukasiewicz. Each satisfies fundamental logical properties such as associativity, commutativity, monotonicity, and identity (Evans and Grefenstette, 2018). However, each t-norm induces different gradient behaviors (van Krieken et al., 2023; Evans and Grefenstette, 2018). In Gödel logic, for instance, the conjunction $\min(x, y)$ passes no gradient to the larger input when $x > y$, which limits optimization dynamics. In Łukasiewicz logic, when $x + y < 1$, the conjunction $\max(0, x + y - 1)$ evaluates to zero and provides no gradient to either input. In contrast, the Product t-norm $(x \cdot y)$ enables gradient flow to both inputs, making it more favorable for learning scenarios. Furthermore, fuzzy logic systems allow for different aggregation operations over constraint dissatisfaction. A common option is the weighted sum, $\sum_{i=1}^{n} w_i s_i$, where $s_i$ is the dissatisfaction score of constraint $i$ and $w_i$ its relative weight.*

**Architecture:** Let $\mathbf{x}_{nn}$ and $\mathbf{x}_{sy}$ denote observed neural and symbolic inputs, respectively, and let $\mathbf{y}$ be the target variables. The output of a neural model $g_{nn}(\mathbf{x}_{nn}; \mathbf{w}_{nn})$ is passed to a symbolic model $\phi_{sy}(\mathbf{y}, \mathbf{x}_{sy}, g_{nn}(\mathbf{x}_{nn}; \mathbf{w}_{nn}); \mathbf{w}_{sy})$ composed of differentiable fuzzy logic constraints.

**Inference:** The inference objective is to predict the most likely output $\mathbf{y}$ by maximizing satisfaction of the symbolic model:

$$\mathbf{y}^* = \arg\max_{\mathbf{y}} \phi_{sy}(\mathbf{y}, \mathbf{x}_{sy}, g_{nn}(\mathbf{x}_{nn}; \mathbf{w}_{nn}); \mathbf{w}_{sy}) = \arg\max_{\mathbf{y}} \sum_{i=1}^{n} w_{sy}^i \phi_{sy,i}(\mathbf{y}, \mathbf{x}_{sy}, g_{nn}(\mathbf{x}_{nn}; \mathbf{w}_{nn})).$$

**Learning:** Learning optimizes both neural and symbolic parameters to minimize constraint violation:

$$\mathbf{w}_{nn}^*, \mathbf{w}_{sy}^* = \arg\min_{\mathbf{w}_{nn}, \mathbf{w}_{sy}} \mathcal{L}(\mathbf{w}_{nn}, \mathbf{w}_{sy}; \mathbf{x}_{nn}, \mathbf{x}_{sy}, \mathbf{y}),$$

where $\mathcal{L}$ quantifies symbolic constraint dissatisfaction.

**Example 8** *Consider the MNIST-Addition task (Appendix B) where the goal is to predict the sum of two digits presented as MNIST images using Gödel fuzzy logic. In this setup, the neural model $g_{nn}$ predicts the truth values of random variables representing the digits, and the Gödel fuzzy logic system enforces constraints regarding their sum.*

*For instance, given two MNIST images, 3 and 5, the target sum is* 8. *A soft logic constraint representing this addition is:*

$$P(x_1)_1 \wedge P(x_2)_2 \rightarrow Sum(x_1, x_2, 8),$$

*where $P(x_1)_1$ and $P(x_2)_2$ represent the truth values of the variables corresponding to the digits output by the neural network and $Sum(x_1, x_2, 8)$ is valid if the sum of the two values sum to eight. Suppose the neural model $g_{nn}$ provides the following outputs:*

$$g_{nn}(3) = \{P(0)_1 = 0.0, P(1)_1 = 0.0, P(2)_1 = 0.0, P(3)_1 = 0.9, P(4)_1 = 0.0,$$
$$P(5)_1 = 0.0, P(6)_1 = 0.0, P(7)_1 = 0.0, P(8)_1 = 0.1, P(9)_1 = 0.0\},$$
$$g_{nn}(5) = \{P(0)_2 = 0.0, P(1)_2 = 0.0, P(2)_2 = 0.0, P(3)_2 = 0.0, P(4)_2 = 0.0,$$
$$P(5)_2 = 0.8, P(6)_2 = 0.0, P(7)_2 = 0.0, P(8)_2 = 0.0, P(9)_2 = 0.2\}.$$

*Gödel fuzzy logic evaluates the satisfaction of the constraint by aggregating the truth values of all valid digit pairs whose sum equals* 8. *Using the fuzzy conjunction operator, the overall satisfaction value is computed as:*

$$\phi_{sy} = \sum_{(i,j)\,:\,i+j=8} \min(P(i)_1, P(j)_2),$$

*where each pair $(i, j)$ corresponds to digits that sum to* 8. *Substituting the neural outputs:*

$$\begin{aligned}
\phi_{sy} = {} & \min(P(0)_1, P(8)_2) + \min(P(1)_1, P(7)_2) + \min(P(2)_1, P(6)_2) + \\
& \min(P(3)_1, P(5)_2) + \min(P(4)_1, P(4)_2) + \min(P(5)_1, P(3)_2) + \\
& \min(P(6)_1, P(2)_2) + \min(P(7)_1, P(1)_2) + \min(P(8)_1, P(0)_2) \\
= {} & \min(0.0, 0.0) + \min(0.0, 0.0) + \min(0.0, 0.0) + \\
& \min(0.9, 0.8) + \min(0.0, 0.0) + \min(0.0, 0.0) + \\
& \min(0.0, 0.0) + \min(0.0, 0.0) + \min(0.1, 0.0) \\
= {} & 0.8.
\end{aligned}$$

## C.2. Neural-Symbolic Probabilistic Logic

A representative example of the *parameter interface* (Section 3.2.3) is found in neural-symbolic probabilistic logic systems, where neural models define the parameters of probabilistic facts (Manhaeve et al., 2021; Xu et al., 2018; Yang et al., 2020). Probabilistic logic extends classical logic by allowing facts and rules to carry uncertainty in the form of probabilities. For instance, the statement "Alice smokes" might be represented with a probability of 0.7, written as 0.7 :: Smokes(Alice). In the NeSy setting, these probabilities are not fixed but are learned from data and predicted by a neural model.

**Architecture:** Let $\mathbf{y}$ denote the target random variables, $\mathbf{x}_{sy}$ the observed symbolic variables, and $\mathbf{x}_{nn}$ the neural observations. Let $g_{nn}(\mathbf{x}_{nn}; \mathbf{w}_{nn})$ be a neural model parameterized by $\mathbf{w}_{nn}$. Its output defines the probabilities of a set of binary facts, which induce a distribution over possible symbolic worlds $w \in \{0, 1\}^n$:

$$p(w \mid g_{nn}(\mathbf{x}_{nn}; \mathbf{w}_{nn})) = \prod_{i=1}^{n} p(w_i \mid g_{nn}(\mathbf{x}_{nn}; \mathbf{w}_{nn})).$$

This factorization follows from the *conditional independence assumption*, where each binary variable $w_i$ is modeled independently given the neural input.

The symbolic model $\phi_{sy}$ encodes logical constraints that are used to filter valid worlds. Each $\phi_{sy}(w) \in \{0, 1\}$ indicates whether world $w$ satisfies the symbolic logic constraints (i.e., $\phi_{sy}(w) = 1$ if the world is consistent with the logic, and 0 otherwise).

**Inference Scenario:** The inference task typically involves computing the probability that the symbolic constraints are satisfied, which corresponds to the *weighted model count*:

$$p(\phi_{sy} = 1 \mid \mathbf{x}_{nn}; \mathbf{w}_{nn}) = \sum_{w \in \{0,1\}^n} p(w \mid g_{nn}(\mathbf{x}_{nn}; \mathbf{w}_{nn})) \cdot \phi_{sy}(w).$$

This sums the probability of all possible worlds $w$ where the symbolic model $\phi_{sy}(w)$ returns true.

**Learning Scenario:** Learning proceeds by minimizing the negative log-likelihood of the symbolic constraint being satisfied, leading to the following optimization objective:

$$\mathcal{L}(\mathbf{w}_{nn}; \mathbf{x}_{nn}) = -\log \sum_{w \in \{0,1\}^n} p(w \mid g_{nn}(\mathbf{x}_{nn}; \mathbf{w}_{nn})) \cdot \phi_{sy}(w).$$

This loss encourages the neural model to assign high probability to worlds that satisfy the symbolic constraints encoded in $\phi_{sy}$.

**Example 9** *Consider the MNIST-Addition example discussed above (Example 7), where the task is to predict the sum of two digits presented as MNIST images. In that example, the model performs an $\arg\max$ to choose values deterministically for NeSy inference (i.e., predicting the sum). However, if the goal is to train the parameters of the digit classifier neural model using probabilistic logics, we can use weighted model counting to compute the probability of the correct sum instead of relying on the non-differentiable $\arg\max$.*

*For instance, given two MNIST images ($\mathbf{3}$, $\mathbf{5}$), with the correct sum as 8. This means that the constraint $\phi_{sy}$ is true only when the sum of the digits in world $w$ equals 8. The images are passed through the neural network $g_{nn}$, which outputs probability distributions*

*over the possible digits for each image:*

$$g_{nn}(\,\mathbf{3}\,) = \{P(0)_1 = 0.0, P(1)_1 = 0.0, P(2)_1 = 0.0, P(3)_1 = 0.9, P(4)_1 = 0.0,$$
$$P(5)_1 = 0.0, P(6)_1 = 0.0, P(7)_1 = 0.0, P(8)_1 = 0.1, P(9)_1 = 0.0\},$$
$$g_{nn}(\,\mathbf{5}\,) = \{P(0)_2 = 0.0, P(1)_2 = 0.0, P(2)_2 = 0.0, P(3)_2 = 0.0, P(4)_2 = 0.0,$$
$$P(5)_2 = 0.8, P(6)_2 = 0.0, P(7)_2 = 0.0, P(8)_2 = 0.0, P(9)_2 = 0.2\}.$$

*Assuming conditional independence between the digits from each image, the weighted model count sums the probabilities of pairs of values that satisfy the constraint $\phi_{sy}$ (i.e., sum to 8):*

$$p(\phi_{sy} = 1; \mathbf{w}_{nn}) = P(0)_1 \cdot P(8)_2 + P(1)_1 \cdot P(7)_2 + P(2)_1 \cdot P(6)_2 + P(3)_1 \cdot P(5)_2 +$$
$$P(4)_1 \cdot P(4)_2 + P(5)_1 \cdot P(3)_2 + P(6)_1 \cdot P(2)_2 + P(7)_1 \cdot P(1)_2 +$$
$$P(8)_1 \cdot P(0)_2.$$

*Substituting the probabilities from the neural network outputs:*

$$p(\phi_{sy} = 1; \mathbf{w}_{nn}) = (0.0 \cdot 0.0) + (0.0 \cdot 0.0) + (0.0 \cdot 0.0) + (0.9 \cdot 0.8) +$$
$$(0.0 \cdot 0.0) + (0.0 \cdot 0.0) + (0.0 \cdot 0.0) + (0.0 \cdot 0.0) +$$
$$(0.1 \cdot 0.0)$$
$$= 0.9 \cdot 0.8 = 0.72.$$

*The loss is then computed as the negative log-likelihood of the constraint being satisfied:*

$$\mathcal{L}(\mathbf{w}_{nn}) = -\log p(\phi_{sy} = 1; \mathbf{w}_{nn})$$
$$= -\log 0.72 \approx 0.33.$$

