# OpenReview forum: "Neural-Symbolic Architectural Axioms of Integration: A Manifesto"
_nesyconf.org/NeSy/2025/Conference_Phase_2 — NeSy 2025 - Phase 2 Poster_

### Official Review · Reviewer_GtLG · 2025-06-29
**A new framework for NeSy**

**Rating:** 2
**Confidence:** 5

**Review:**

The authors propose a set of “architectural axioms” for neurosymbolic integration which neurosymbolic systems can be combined.  They identify two large categories: catalyst-based and interface-based.  The former characterizes transformations across boundaries while the later characterizes how the neural and symbolic systems work together.

As the authors point out, there are many existing ways, they do not mention one of the most important frameworks introduced by Henry Kautz at a keynote at AAAI.  The keynote itself is widely referenced (including by some of the papers the authors cite) and the subsequent AI Magazine Article has over 200 citations – so it seems a significant oversight to not mention this paper.  It is also surprising that the authors do not mention ideas from the cognitive science literature (e.g., dual process theory; System 1 / System 2) – especially as Nobel Laureate psychologist Daniel Kahneman is frequently cited by researchers in the neurosymbolic literature.  Further, another important framework (van Bekkum, cited by the authors) is somewhat mis-characterized in the paper.  The van Bekkum paper does more than provides visual metaphors for classifying NeSy systems (as the authors state), but also introduces elementary and compositional patterns (derived from primitives) that capture many of the ideas introduced in the submitted paper.

It seems the major contribution is that the authors created the larger categories, as the examples in each seem to be taxonomic categories already introduced by Kautz or the van Bekkum paper cited by the authors.  However, the authors only provide examples of frameworks of the axioms; it is not clear what advantage is obtained or what new insights are gained by employing their proposed strategy.  In fact, some things remain rather unclear.  For example, if you induce rules from a neural network training process (like dILP) that seems to be considered symbolic extraction – where the neural side catalyzes the symbolic side.  Meanwhile, a generated logic program by Logic-LM, where an LLM generates a logic program is considered “structure interface” where the LLM interfaces with a solver.  However, dILP still needs a solver while Logic-LM is still generating a logic program that catalyzes the logic.  Fundamentally, in both cases the neural approach is generating a logic program, it’s hard to see how these are different unless we say one is LLM-specific for some reason (but the authors repeatedly say this was not meant to be system-specific).  There are similar issues with other paradigms.

On the whole, due to prior work like Kautz and van Bekkum, there is a high bar for a new foundational framework for NeSy, and I do not believe this paper meets that bar in terms of novelty or a framework that has the capability to provide new insights to the field.

**Anonymity:**

Remain anonymous

---

### Official Review · Reviewer_sZUV · 2025-07-08
**Neural-Symbolic Architectural Axioms of Integration: A Manifesto**

**Rating:** 7
**Confidence:** 5

**Review:**

The paper proposes a set of architectural axioms to provide a unified theoretical foundation for neural-symbolic (NeSy) AI, a field that integrates neural networks (connectionist) and symbolic reasoning to achieve interpretable, robust, and generalizable AI systems.

Comments:
1. The lack of a clear mapping between existing systems and the axioms weakens the claim that they reduce fragmentation.
2. Including case studies (beyond MNIST-Addition) or a qualitative analysis of how existing systems map to the axioms would strengthen the paper’s practical relevance.
3. The lack of empirical analysis undermines the axioms’ utility. Please add a case study with metrics like accuracy or inference time to validate their effectiveness against neural or symbolic baselines.

**Anonymity:**

Remain anonymous

---

### Official Review · Reviewer_8QmX · 2025-07-08
**The authors present an axiomatic framework for neural-symbolic integration. Their main point is that box-and-arrow designs can be systematically represented as families of principles. Catalyst axioms, which describe how one modality constructs or extracts structure from the other, and Interface axioms, which define the loci and mechanics of ongoing interaction (variables, parameters, structures). By abstracting away from learning algorithms or inference engines, the authors aim to give the NeSy community a common, implementation-agnostic vocabulary for comparing and composing hybrid systems. The idea is very compelling, but the execution lacks formal precision and empirical validation.**

**Rating:** 6
**Confidence:** 2

**Review:**

Strengths:
1. There is an ambitious unifying vision.
2. The definitions of the top-level taxonomy are very clear. The splitting of integration mechanisms into transformation-based Catalysts and interaction-based Interfaces neatly separates where and how symbolic and sub-symbolic components meet.
3. The conclusions very clearly highlight the need for work such as that by the authors. The narrative as a whole is well grounded in theory, and the examples are good representations of the logics and methods they wish to build an axiomatic framework for.


Weaknesses:
1. The Programmatic Catalyst category is so broad that standard CNNs or Transformers qualify. This makes it difficult to draw the line between “neural architectures” and “neuro-symbolic integrations.”
2. There isn’t an empirical validation of the utility of this taxonomy. While it is important for the community to have a well-defined vocabulary, beyond this, the authors present no practical utility of their proposed taxonomy.
3. There is almost no attention paid to the complexity or undecidability that arises from the different inferences explored.
4.Considering the current landscape of large language models, the authors bring up prompt-engineering and in-context reasoning as open questions but do not bring them into the fold of their taxonomy. This leaves their work vulnerable to obsolescence.

**Anonymity:**

Remain anonymous